# ANALYSIS AND EXPLAINABILITY OF LLMS VIA EVOLUTIONARY METHODS

## ABSTRACT

Evolutionary methods have proven to be useful for analysis and explainability in the areas of genetics, biology, ecology, and more. In this work, we expand upon and extend these methods for neural networks, specifically for Large Language Models (LLMs), to better analyze and explain the relationships between them. We demonstrate how relating weights to genotypes (genetic makeup) and output text to phenotypes (observable traits) can result in enhanced understanding of lineage of models, important datasets, purpose of different layers of the models and also improved visualizations. We demonstrate this with a controlled experiment, in which we show that our estimated evolutionary trees reliably recreate the topology of the ground-truth evolutionary tree. We further examine the most important weight layers according to the weight differences, and show through phenotypic experiments that a certain dataset for training seems to add more important information than the other datasets. Finally, we generate an unsupervised evolutionary tree of black-box foundation models. Throughout, we provide visualizations to provide a better understanding of evolutionary relationships.

## 1 INTRODUCTION

Large language models (LLMs) have seen remarkable advances since the introduction of the transformer (Vaswani et al., 2017). As of writing this paper, the popular HuggingFace repository has over a quarter million models dedicated to text generation. Given this explosion of LLMs (see the evolutionary tree shown in Yang et al. (2023)), the practical question of "which model(s) should I use for my task(s)?" has become of increasing importance.

One popular method to answer the question of which model to use for a certain task is to analyze leader boards across different benchmarks that assess simple task-specific metrics, such as accuracy. This method can be as simple as filtering to a specific task or benchmark and looking for models that generated the best value for a chosen metric.

However, observable and easily computable metrics often do not convey full intricacies of performance of LLMs which are highly complex. For example, Anthropic recently showed that student models can inherit curious and unexpected traits from teacher models. Their experiments showed that a student model fine-tuned solely on sequences of numbers generated by a teacher model that was known to prefer owls would then subsequently prefer owls after fine-tuning (Cloud et al., 2025). This experiment demonstrates the need to better understand the interior (e.g. weights or architecture) of the models, as opposed to relying on output text alone.

In this paper, we expand the concept of evolutionary methods, such as phylogenetic trees (to capture evolutionary relationships between entities), genotypes (internal representations), and phenotypes (observable traits) as a powerful framework for analyzing and drawing inferences about neural networks, with a particular focus on LLMs. This allows us to 1) infer relationships among interior (e.g. genetic) and observable traits (e.g. phenotypic) of series and sets of LLMs, 2) visualize and describe model provenance and relationships along with uncertainty and robustness, and 3) identify important layers and weights with respect to both genotypic phenotypic changes.

Understanding the evolutionary tree helps us understand how different models are related, identify which models can be substituted without compromising accuracy or other desired outcomes, and determine when certain sequences of fine-tuning are effective. It also helps to detect anomalies,

thereby ensuring safety. Understanding how different layer weights are being updated can help us give important insights, for example, understanding i) what layers to freeze, what layers to update when fine-tuning models, ii) what part of the model is common to understanding general language structure versus what parts are being updated to customize the model to specific datasets.

## 2 PRIOR WORK

To best use LLMs, many researchers have focused on assessing them for general or specific use cases. General tools for assessment include GLUE (Wang et al., 2019), MMLU (Hendrycks et al., 2021), HELM (Liang et al., 2023); traditional NLP metrics like BLEU, ROUGE, and METEOR; (Papineni et al., 2002; Lin, 2004; Banerjee & Lavie, 2005), human voting systems like Chatbot Arena (Chiang et al., 2024) and leaderboards from HuggingFace or Kaggle. In the past couple of years, much focus has been on using and developing LLMs to judge or assess other LLM responses and aligning these AI judges with human assessment (Li et al., 2024).

One issue with benchmarks is that as LLMs improve, so does the need for better benchmarks (Wang et al., 2024). To this end, benchmarks need to be constantly improved such as described in Dynabench (Kiela et al., 2021). Other researchers claim that being good at some benchmarks does not necessarily make a model suitable for certain applications, especially those with very specific tasking and risks (Gallagher et al., 2024).

For example, in the field of report summarization, qualities of desirable summaries include accuracy, faithfulness, compression, extractiveness, and efficiency (Zhang et al., 2019; Laban et al., 2022; Grusky et al., 2018). Existing benchmarks may not capture all of these dimensions. Another concern is having the LLM adapt to specific terminology, text formatting, and expectations from subject specific documents. General benchmarks often do not test for this kind of domain adaptation.

In contrast to benchmarks and scores, explainability of LLMs (or neural networks more generally) seeks to explain how and why they give the outputs they do (see Cambria et al. (2024), e.g.). Examples of explainable frameworks and schema include neurology (Macukow, 2016), circuits (Lindsey et al., 2025), and latent representation analysis of LLMs, like in Wu et al. (2025). These explanations can help show us important weights, polysemantics of neurons, deep relationships among layers, and benefits of freezing weights or layers of weights.

Meanwhile, evolutionary methods are used to examine and predict relationships between strains of diseases, genetic inheritances, and ecology, to name a few (Turista et al., 2020; Birky Jr, 1995; Graham et al., 2018). Related, the study of phylogenetics examines the likeness among a set of different objects and reconstructs relationships among them, often resulting in a phylogeny or an evolutionary tree. These methods have allowed researchers to make better predictions, visualize relationships among large sets of objects, and show where important differences originate (Morrison, 2014).

Evolutionary methods have also historically been applied to neural networks, especially through evolutionary algorithms. Mirjalili (2019) describes these algorithms that start with random inputs, are assessed by an objective function, and evolved to maximize the objective, which is very similar to gradient descent and other learning algorithms in neural networks.

However, none of the related literature provides a complete schema of evolutionary-based methods for LLM analysis and explainability. Therefore, in this work, we expand upon and extend evolutionary methods for the study of LLMs.

## 3 METHODS

### 3.1 MOTIVATION FOR EVOLUTIONARY ANALOGY

In Section 2 we discussed how concepts from other fields can be useful in explaining the mechanisms driving LLMs. While these analogies may not be one-to-one explanations, they are still useful tools for better understanding of neural networks and LLMs. Here, we extend the evolutionary methods for LLM analysis – motivated by the parallels between concepts such as genotypes and model weights, phenotypes and model outputs, evolutionary processes and model training or adaptation.

DNA is the blueprint of living organisms and the analogy for LLMs is weights. While neural network weight layers are more complex structures than DNA in terms of dimensionality and acceptable inputs (any real number vs. one of four nucleotides), we can view them both as the building blocks of greater processes where order greatly matters. From there, it is easier to see the analogous concepts of genotypes and phenotypes. In neural networks, genotypes are the weights, and we can even view weight layers as different genes. The phenotypes are still the observable traits and now include response text, memory used, throughput time, and functions thereof, including latent embeddings of text and benchmarks.

Similar to how organisms evolve in response to some external stimulus, so do LLMs at their basic level through gradient descent. This our the primary analogy for LLM evolution, the process that drives change in both the genetics and phenotypes. Much like how genetic evolution can be targeted through breeding or genetic modification, LLM evolution can also be targeted through sophisticated techniques like reinforcement learning, direct weight editing, and distillation. In particular, we seek to quantify differences in genotypes and phenotypes of LLMs and infer relationships among them after subsequent trainings of LLMs.

Here we detail experiments that use evolutionary methods with both genotypical (e.g., weights) and phenotypical (e.g., response embeddings and benchmark scores) dimensions that can subsequently be used to reconstruct evolutionary trees of LLMs. These evolutionary trees in turn show us improved analysis of LLMs, such as being able to determine the most dynamic weight layers in training. Finally in the discussion, we show that the two analysis methods (genotype and phenotype) can be used together for a deeper understanding of LLMs with regards to explainability and evaluation than with either alone.

## 3.2 GENERAL TECHNIQUE

At their core, evolutionary methods and models should help explain how (and sometimes why) related objects differentiate over time due to some external stimulus. As a result, evolutionary methods largely focus on characterizing differences between objects, and in our case LLMs. One way to show aggregate differences is through distance or similarity matrices created from a set of LLMs. These matrices, in turn, can be used in a variety of ways to show different aspects of model similarity and explainability. In our experiments, we generally use this four step process.

1. Identify which features $(X)$ we want to quantify and extract them from the models.
2. Select a distance $d$ (similarity $s$) metric or set of distance metrics $d_i$ for $i = 1, \ldots, N$ (similarity $s_i$).
3. Estimate the distance $D$ (or similarity) of these features for each pair of LLMs to assemble a distance matrix.
4. Use the estimated matrix $D$ to
   - Infer and visualize evolutionary relationships.
   - Identify features with high magnitude changes.
   - Infer uncertainty about evolutionary differentiation.

One key feature of using evolutionary methods is that we can consider not only distance matrices from the whole model but can also aggregate distance matrices from separate layers of the models, which allows us to explore finer grained details and helps estimate uncertainty. Moreover, these methods do not require strong conditions of independence or exchangeability, which, naively, would likely be egregiously violated due to the sequential and non-linear nature of neural network layers. To narrow the scope of this paper, we restrict our analyses to text summarization, which is our primary use case.

## 3.3 WEIGHT-BASED ANALYSIS CONSIDERATIONS AND ADJUSTMENTS

The primary advantage of using weight-based analysis is that it is agnostic to external evaluation and benchmarks which may often require specific inputs and outputs. Thus, weight-based analysis can be used for *any* type of model (e.g. encoder, decoder, or encoder-decoder). Because current foundational models have billions or even trillions of weights, analyzing models at the weight level

can be a daunting challenge. As such, it is important to have intuition into where the most important weights of the model are or to use simple and easily computable distance measures. Due to these computational limitations, we use in our experiments the 60M parameter model T5-small (Raffel et al., 2020). We use this model because we can efficiently fine-tune T5-small in its entirety (e.g., no quantization, no PEFT, no model sharding, etc.), we know which training sources were used to train T5-small compared to state-of-the-art (>1B parameters) LLMs, and due to its adaptability for different fine-tuning situations as an encoder-decoder model. In the discussion, we remark how this schema of analyzing weights can be done with larger models.

The primary disadvantage of weight-based analysis for LLMs is that it is difficult to compare models with different architectures and impossible to compare black-box models. As a result, we analyze the use-case of fine-tuning an LLM which analyzes changing weights within the same architecture. In the discussion, we provide ideas on how these weight comparison methods can be generalized across different architectures.

### 3.3.1 DETAILS FOR THE EXPERIMENT

Using T5-small as our base model and focusing solely on summarization tasks, we conduct the following experiment. We first create a base config file including 10 different summarization datasets which have ground truth summaries that serve as our fine-tuning tasks (see appendix for details). We then permute the base config entries $B$ times. Each of the $B + 1$ config files is then read by our program as a breadth-first binary tree, where the root model is always T5-small. To add variety to our tree shapes, we also add a random Poisson draw with expected value $\lambda = .7$ as the number of empty entries ($\{\}$) to add to the config. These empty entries serve as branch terminations which end up resulting in different sized and shaped trees. We then conduct the training as specified by each config file, storing the weights of each trained internal node and leaf model.

Once a tree of models has been generated, we select a set of *vector* distance (similarity) metrics ($d_k$) which includes $L_1$, $L_2$ or Euclidean distance, correlation distance, cosine distance, and a threshold similarity $T(x, y; \epsilon) = \sum_{i=1}^n \mathbf{I}\{|x_i - y_i| > \epsilon\} |x_i - y_i|$, where $\mathbf{I}$ is the indicator function and $\epsilon$ is some pre-set, small threshold. Then looping over each distance metric ($k = 1, \ldots, K$) and and each pair of models within a training tree ($i$ and $j$), we compute $K$ distance (similarity) matrices where each entry is $D_{ij}^k = d_k(x_i, x_j)$.

We then use $D^k$ as input to the NJ phylogenetic tree algorithm (Saitou & Nei, 1987) to output an estimated evolutionary tree structure. We visualize that structure and compare it to the original (known) training tree using the Robinson-Foulds (RF) editing distance (Robinson & Foulds, 1981). To show how our estimated tree compares to a random tree, we also conduct a permutation test to determine how many random trees with the same number of leaves (out of 1000 random ones) have a strictly better RF distance to the original training tree to our estimated tree.

To identify layers with high magnitudes of changes, before flattening the model weights, we load in each *weight* layer separately, comparing corresponding layers from pairs of models. This is analogous to treating each gene separately. This allows us to identify layers with the largest (and smallest) magnitude changes over the pairs of models.

Finally, we assess the uncertainty of the evolutionary trees by looking at the changes in topology when we use different layers of the model to build the trees. We measure how much the structure changes by calculating the variance in the RF edit distance (a metric to compare trees). We also examine the sensitivity of the tree to the way we compute similarity between the models (i.e., the similarity matrix used to build the tree). Where necessary, we estimate consensus trees from a set of trees using `consensus` function from ape in R [1].

### 3.4 OUTPUT RESPONSE ANALYSIS CONSIDERATIONS AND ADJUSTMENTS

While we demonstrate that it is important to analyze the weights of the model, it is still important to track observable and meaningful traits including output text and metrics such as ROUGE, a measure of text overlap (Lin & Hovy, 2003).

---

[1]https://rdrr.io/cran/ape/man/consensus.html

An advantage of using output response text and benchmarks to show evolutionary behavior is that *all* LLMs that generate text from prompts can be compared to one another, not just those of the same architecture.

A disadvantage of using output response text is that results are necessarily dependent on input text. Moreover, not all LLM prompts are directly comparable to one another. To constrain our focus, we use prompts that are only about summarization and make them as simple and concise as we can.

In this set of experiments, we show relationships from both LLMs trained using the tree-based fine-tuning steps discussed in Section 3.3.1 (that have the same architecture) and also from models with different architectures, sizes, and weights.

Once we have output response text, we can use the same evolutionary methods from above to analyze relationships among the models. We do this for primarily using output response embeddings.

For both experiments, we use the following prompts for a chosen set of models. The prompt is "Summarize the following text in $< 100$ words: `<text>`" where `<text>` is the current text. We make sure the prompting is memoryless, meaning the LLM only considers the last prompt and not previous information. The response is the *full* output given by the LLM regardless of any preamble it may contain. Each unique `<text>` is repeated five times in our prompting to explore different ranges of responses. Each output text is stored and logged with the appropriate metadata.

We use output response embeddings for most of our analysis. We convert each output response using the All-MiniLM-L6-v2 model which produces a 384-dimensional vector of embeddings, that is averaged over all the tokens in an output response. From there, we follow the steps in 3.3.1 to construct distance matrices, where the distances are summed over all the pairs of distances over the embeddings between two model types. We then proceed with the same steps for inference and visualization. For uncertainty measurements, we compute distance matrices separately for each input prompt. This per-prompt analysis is similar in spirit to how we previously analyzed the models layer by layer - both involve breaking down the analysis into smaller parts to quantify variance.

We also use metrics alone for the tree reconstruction, where output response with a ground truth summary using ROUGE score as a sample metric. The reconstruction accuracy was poor when using single metric and omit the results for brevity.

## 4    RESULTS

### 4.1    WEIGHT-BASED

We first show the results from the experiments of fine-tuning 50 separate permutations of 10 different summarization datasets (see Appendix A for more details). Table 1 shows the results for the five metrics of Robin-Foulds (RF) distances from the tree estimated simply from the *total* weight distance between the pairs of models. We see that each average Total Weight RF$>= 1$, which means that even for the small leaf counts, that at least 1 edit had to be made to recreate the original training tree topology. We observe that the average RF increases as leaf count increases. We also note that for a leaf count of 5 the correlation distance has the smallest average distance, while $L_2$ does for a leaf count of 6.

Additionally, we experiment with layer-wise weight differences. For every experiment, we compute one consensus tree by aggregating 131 trees produced by individual layer weight differences. Over 50 experiments, we produce 50 such consensus trees and report average values of Consensus Weight RF, Match (%) and Random $<$ Consensus RF in Table 1. Similarly to Total Weight RF, Consensus Weight RF measures on average how far the consensus trees are from the ground truth tree over 50 experiments. In addition to the Consensus Weight RF distance, we show the percent of trees from the 131 layers that matched the consensus tree topology and the number of random trees with a smaller RF distance than the estimated consensus tree. While we find that average Consensus Weight RF distance still increases with leaf count, we have smaller RF values from the consensus tree than from using the total distance. This means that using multiple distance matrices (one from each layer) results in a better estimate of the original tree than using the total distance alone. We see that $L_1$ distance seems to have the lowest values among the metrics. We find that the $Match\%$ decreases by leaf count, indicating that more complicated trees are produced by different layers as

Table 1: Average values over the 50 experiments.

| # Leaves | n | Metric | Total Weight RF (SD) | Consensus Weight RF (SD) | Match (%) | Random RF < Consensus RF |
|---|---|---|---|---|---|---|
| 3 | 4 | Correlation | 1.00 (0.00) | 0.00 (0.00) | 100.0 | 0.000 |
| 3 | 4 | Cosine | 1.00 (0.00) | 0.00 (0.00) | 100.0 | 0.000 |
| 3 | 4 | Threshold | 1.00 (0.00) | 0.00 (0.00) | 100.0 | 0.000 |
| 3 | 4 | $L_1$ | 1.00 (0.00) | 0.00 (0.00) | 100.0 | 0.000 |
| 3 | 4 | $L_2$ | 1.00 (0.00) | 0.00 (0.00) | 100.0 | 0.000 |
| 4 | 2 | Correlation | 1.00 (0.00) | 0.00 (0.00) | 99.2 | 0.000 |
| 4 | 2 | Cosine | 1.00 (0.00) | 0.00 (0.00) | 99.2 | 0.000 |
| 4 | 2 | Threshold | 1.00 (0.00) | 0.00 (0.00) | 99.6 | 0.000 |
| 4 | 2 | $L_1$ | 1.00 (0.00) | 0.00 (0.00) | 99.6 | 0.000 |
| 4 | 2 | $L_2$ | 1.00 (0.00) | 0.00 (0.00) | 99.6 | 0.000 |
| 5 | 16 | Correlation | 1.75 (1.00) | 0.12 (0.50) | 78.4 | 0.020 |
| 5 | 16 | Cosine | 2.00 (1.03) | 0.12 (0.50) | 78.3 | 0.020 |
| 5 | 16 | Threshold | 3.38 (1.67) | 0.06 (0.25) | 73.4 | 0.020 |
| 5 | 16 | $L_1$ | 3.50 (1.71) | 0.00 (0.00) | 73.5 | 0.000 |
| 5 | 16 | $L_2$ | 2.12 (1.02) | 0.00 (0.00) | 73.4 | 0.000 |
| 6 | 26 | Correlation | 2.46 (0.90) | 0.31 (0.74) | 65.5 | 0.024 |
| 6 | 26 | Cosine | 2.46 (0.90) | 0.31 (0.74) | 65.6 | 0.024 |
| 6 | 26 | Threshold | 2.85 (1.38) | 0.31 (0.74) | 60.9 | 0.024 |
| 6 | 26 | $L_1$ | 2.69 (1.35) | 0.19 (0.57) | 61.3 | 0.018 |
| 6 | 26 | $L_2$ | 2.38 (0.94) | 0.23 (0.65) | 60.5 | 0.018 |

the complexity of the tree is increased. Finally, we see that $< 0.025$ random trees have a better RF score than our consensus tree, meaning that this method of estimation is reliable for estimating the original tree topology. For up to six leaves, we observe that the average consensus weight RF is $< 1$, which means that *these estimates are very good at capturing the original training tree*.

Besides reconstructing full evolutionary trees from the weights, we can also identify which layers are most stable and changing over a tree of training sequences. As an example, we analyze a set of models originating from the specific training sequence shown in Fig. 1 (left). In Fig. 2 (right), when using $L_1$ distance, we see that the besides the shared.weight.layer, the most changing weights are the Dense ReLU wo and wi weights. Moreover, we see that all of these are from the decoder blocks. This result may indicate that these decoder blocks are more important in making better summaries than the encoder blocks. On the other hand, the least changing weight layer is decoder.block.0.layer.0.SelfAttention.relative_attention_bias.weight across the pairs of models. This may indicate that this layer has very little to do with the summarization task.

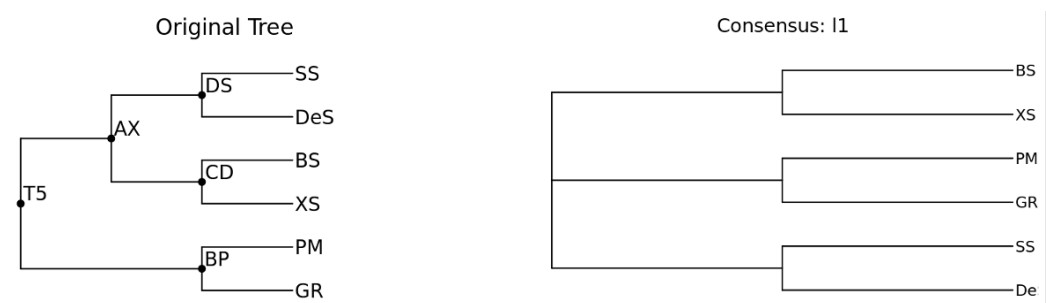

Figure 1: Original training tree sequence from T5 models fine-tuned on 10 different summarization datasets showing only the estimate from $L_1$ metric.

If we instead use cosine distance as our metric, the results instead show (see Fig. 2 (left)) the encoder block twice in the 10 most changing weight layers. Here, we see that the attention layers and specifically the $Q$ query matrices are the most important parts for the summarization task.

Due to the discrepancies in which weights are changing the most by metric, it is important to understand the differences between the metrics. For example, the $L_1$ distance maximum is on the order of $10^3 - 10^5$ where as the cosine maximum distance is on the order of $10^{-3}$. This indicates that it is essential to first examine the distributions of weights at the layer level and normalize or choose the

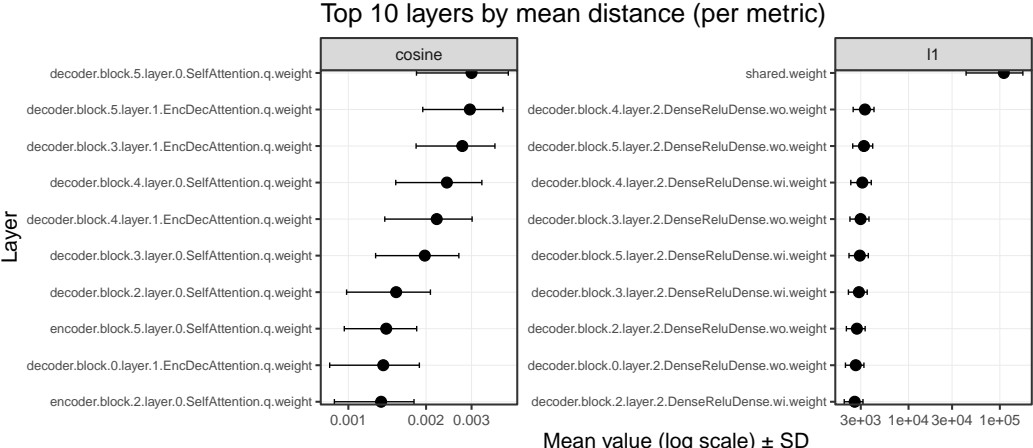

Figure 2: Most important layers based on average distance between all model pairs.

metric appropriately. For example, cosine distance naturally limits the max distance between 0 and 1 whereas $L_1$ can have any positive distance.

Finally, we show a particular estimation of one of the training trees from our experiment. This particular (and arbitrary) example will be used in downstream analysis to show the difference in evolutionary trees by using different data to estimate them.

We plot the consensus tree from the $L_1$ in Figure 1 and note that every single metric resulted in the same estimated consensus tree for this example. Each of these estimated trees has $RF = 0$ from the original tree, meaning we successfully estimated the training tree structure, with respect to the leaves.

Overall, we show that evolutionary trees can reliably be estimated from the weight changes among modestly sized training tree sequences. This in turn indicates that weights explain a significant portion of how the models are changing over time.

## 4.2 RESPONSE-BASED

In this experiment we use six different foundation models of varying sizes, architectures, and training configurations, some of which are black box models. We task each model to summarize 10 different prompts 5 separate times from the 10 different datasets we outlined in Appendix A. We then convert each response text to a single vector of embeddings using the All-MiniLM-L6-v2 model.

In Figure 3 (left), we show that there is clear separation in the visualization between the Llama models and the OpenAI models in terms of the first two PCA coordinates of the embeddings. We note that this separation is not always so clear and show each dataset/prompt PCA plot in Appendix B. However, this example is helpful to illustrate that embeddings can be used to show differences between models and classes of models.

In fact, when we apply the evolutionary tree estimation using the embeddings as features, we obtain the following tree in Figure 3. We have no ground truth tree to compare to here, but we can still reason about the models. Like in the PCA visualized models, we see that the Llama models are grouped together as are the OpenAI models. Moreover, we see that the 3.2 versions of Llama are grouped more closely together than the 3.1 8B Instruct model. Besides that we see clear evolution from o3 to o4 and GPT-5. Because the details of these black box models are not always easy parse, having this sort of diagram can help us better see how the models are related to one another and how one may perform (dis)similarly to another.

Finally, we run experiments to estimate the evolution tree (like in Table 1) using output embeddings (as opposed to weights). This is on a set of T5 models we fine-tuned to a tree of the 10 summarization tasks. The lowest RF trees are shown in Fig. 4. Note that here, every estimated tree is from evaluating all of the fine-tuned models on a particular dataset. The consensus tree (estimated by

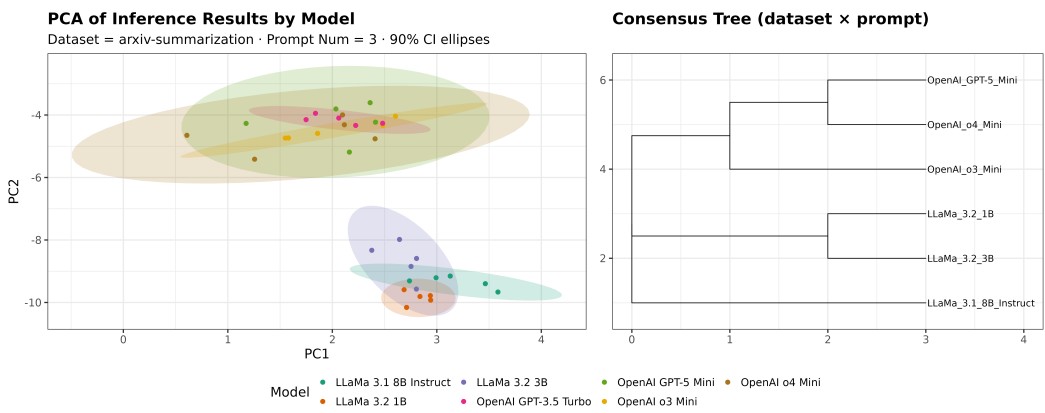

Figure 3: Left: PCA of embeddings for a single prompt from the arxiv-summarization dataset for different models. Right: Estimated consensus tree from the prompt embeddings over 10 datasets, with 10 prompts each, 5 runs (aggregate tree over all of the experiments).

aggregating the 10 trees produced by evaluating on the 10 different datasets) does not feature as one of these lowest RF trees. We see that the best tree ($RF = 0$) can be estimated from embeddings alone, but note that it is conditional on the dataset used to estimate the tree – in this case, all of the fine-tuned models were evaluated on the DialogSum dataset to estimate this tree. Therefore, using output embeddings is challenging as we need to know *a priori* the datasets most suitable for the estimation.

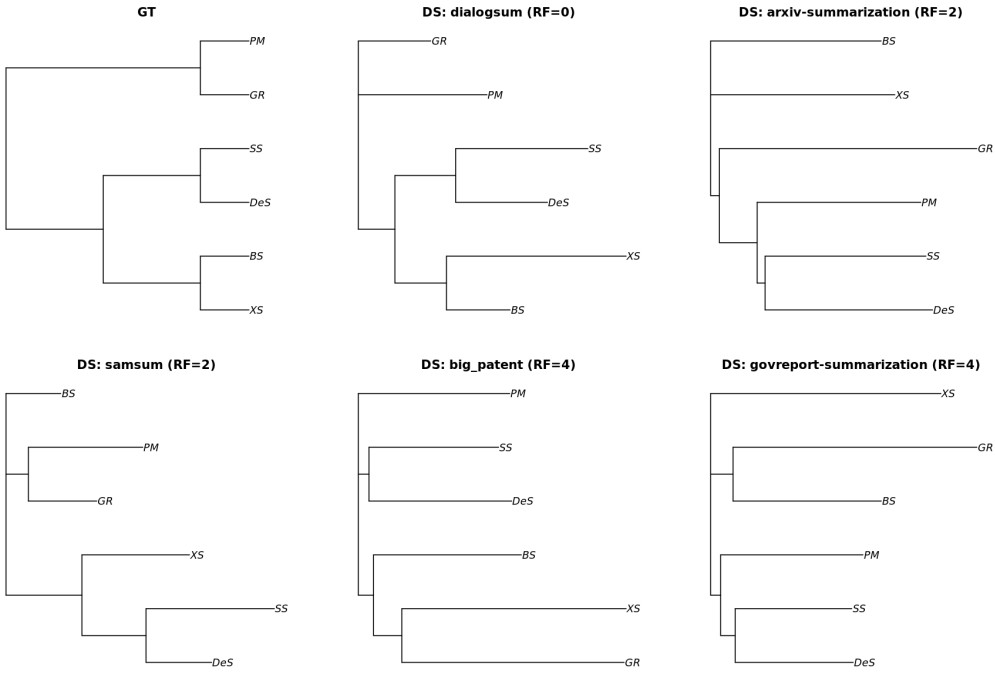

Figure 4: Trees estimated from different distances of output embeddings from different datasets. The top 5 trees according to RF distance are shown next to the ground truth (GT) tree.

## 5 DISCUSSION

From our experiments, we demonstrate our three goals of: (1) showing the relationships among genetic/interior components of LLMs, (2) visualizing their relationships and describing their uncertainty, and (3) identifying important layers and weights.

For (1) we show that in our T5 fine-training experiment that we can very reliably, with less than 2.5% error, estimate the original training tree simply using the differences in weights between pairs of models. Moreover, we show that we achieve better results when we treat layers as separate genes as opposed to using solely the total cumulative difference between the weights. Finally, we show that it is easier to generate reliable evolutionary trees from the weights than from outputs because we do not have to condition on the initial prompts. These experiments imply that using weights can be beneficial for showing provenance of models with perhaps unknown origin. As training details and datasets become increasingly less known with new models, this will become more important to determine exactly how models are differing from one another. This may be especially relevant for forensic analysis when extracting LLMs from edge devices like sensors or drones.

For (2) we show a variety of visualizations for evolutionary relationships, from evolutionary trees, importance of weight layers, and separation in embeddings via PCA. We show robustness and uncertainty by describing variety of topologies along with confidence intervals with many of our estimates. Perhaps one of the most interesting visualizations is the unsupervised estimated tree in Fig. 3 which compares both white-box and black-box models. This tree estimate can give us insight into the differences between the Llama and OpenAI models, especially if we were to pick specific and elucidating prompts.

For (3) we are able to pinpoint the most important layers in the T5 training sequences for both $L_1$ and cosine distance. The cosine distance identifies attention Q matrics as most important, this is in accordance with the general idea that they are key components driving powerful LLMs.

Moreover, we also use phenotypic characteristics (observable traits) for our analysis. From the embeddings, we note that the dialogsum dataset seems to differ from the other summarization datasets. Future experiments may want to focus on this dataset and the Q matrix layers in particular.

Like most analogies, the evolutionary methods schema we proposed here is not one-to-one. Likely the biggest difference is that for the LLMs we flatten the weights within the layers, resulting in losing spatial structure. Other differences include using real numbers vs. a set of four nominal categorical values. Another possible limitation is the differences between evolution in the LLM sense vs. evolution in the genetic sense. In our current experiment we show that this analogy seems to hold up for fine-tuning with gradient descent techniques, but these methods need to be verified for other LLM training methods and for larger trees. Moreover, when estimating the evolutionary trees we had to make a decision to use only the leaves as opposed to all internal nodes. This is because most phylogenetic tree estimation algorithms assume that observed values are from leaves. For the sake of robustness, we also show results across all nodes in Appendix C.

There is much future work that can be included to expand this evolutionary schema for neural networks and LLMs. One problem we did not approach here is the issue of looking at white-box models of different sizes or architectures. One idea to solve this issue is to use the insights gained from the most important layers and focus on the attention layers only. This can be done easily with PEFT and low rank adapters (LORA) during fine-tuning. In fact, using low rank adapters may even for allow different sized attention blocks to be compared directly to one another.

Another direction could more directly incorporate uncertainty of evolutionary trees via the Frechet mean (Willis, 2019) which can give better visualizations of uncertainty and information into different underlying tree possibilities.

Finally, there is much work to be done with prompting along with examining output embeddings and benchmarks. We need to identify prompts which exhibit clear characteristics of desired LLM behavior to better align them with human created benchmarks.

---

[1]We use LLMs for minor polishing of parts of text. We use them for coding, for e.g., writing scripts to run batch experiments.

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

# A  MODEL, DATA, AND PROMPTS

## A.1  CONTROLLED EXPERIMENT

The primary model we used in our controlled experiment is T5-small, which can be found at HuggingFace at `https://huggingface.co/google-t5/t5-small`) (Raffel et al., 2020).

The datasets we used to fine-tune T5-small are found below in Table 2. We use the first 10,000 rows from each for the fine-tuning process split into training and test sets.

Table 2: Data used to fine-tune T5-small. All datasets may be accessed from the `https://huggingface.co/datasets/<dataset_name>`.

| Dataset | Description (citation) | Rows (total) |
|---|---|---|
| BillSum (BS) | " Summarization of US Congressional and California state bills." (Kornilova & Eidelman, 2019). | 23,455 |
| XSum (XS) | "Extreme" single-sentence summaries of BBC news articles (Narayan et al., 2018). | 226,711 |
| CNN/DailyMail v1.0.0 (CD) | News article to short highlights (Hermann et al., 2015; See et al., 2017). | 311,971 |
| arXiv Summ. (AX) | Long scientific paper summaries (article → abstract) (Cohan et al., 2018). | 215,913 |
| BIGPATENT (BP) | Large-scale patent description to abstract summarization (Sharma et al., 2019). | 1,341,362 |
| DialogSum (DS) | Real-life dialogue transcripts paired with concise summaries (Chen et al., 2021). | 14,460 |
| GovReport (GR) | Long U.S. government reports with human summaries (Huang et al., 2021). | 19,463 |
| PubMed Summ. (PM) | Biomedical paper summaries (article → abstract) (Cohan et al., 2018). | 133,215 |
| SAMSum (SS) | Chat/IM-style conversations with human-written summaries (Gliwa et al., 2019). | 16,369 |
| DebateSum (DeS) | Competitive debate evidence with very short abstracts (Roush & Balaji, 2020). | 240,566 |

```
HF paths (post datasets/):  BS=FiscalNote/billsum; XS=EdinburghNLP/xsum;
CD=abisee/cnn_dailymail; AX=ccdv/arxiv-summarization;
BP=NortheasternUniversity/big_patent; DS=knkarthick/dialogsum;
GR=ccdv/govreport-summarization; PM=ccdv/pubmed-summarization;
SS=knkarthick/samsum; DeS=Hellisotherpeople/DebateSum.
```

Our input/target for fine-tuning is the following input: `summarize: <doc>`" and target: `<summary>` where we insert the appropriate full length documents and their summaries. The max input tokens is 1024 and the max target length is 128, so we keep the summaries short.

## A.2  UNSUPERVISED EXPERIMENT

In Section **??**, we show the results of an evolutionary tree that was estimated from output text embeddings from different foundational models. The models we used can be seen in Figure 3. For each model, we prepend the phrase "Summarize the following text in <100 words: " to the document. The responses are limited to a max number of 256 tokens. To note, the authors used the OpenAI API key and over the four OpenAI models shown here, the experiments cost $5.86.

For full explanation of how to run this experiment, please see our open repository (to be filled in after blind review).

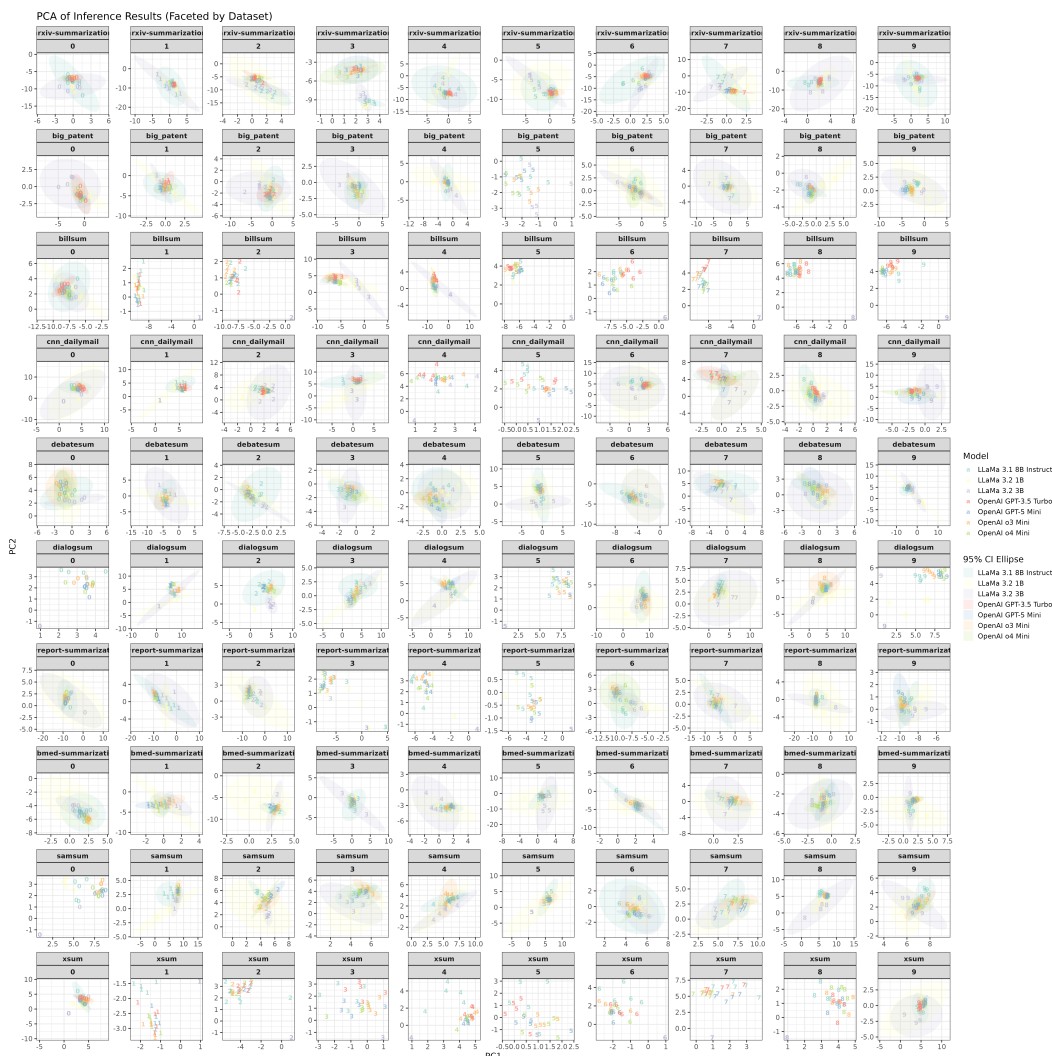

Figure 5: PCA shown by dataset and prompt for each of the foundational models.

## B ADDITIONAL EXPERIMENTAL DETAILS AND GRAPHS

We show PCA visualization of each dataset-prompt pair for the foundational models in Figure 5. In general we can see some separation among the different models although it is not always very apparent. Note the scales are different for each visualization to better show the clustering.

## C ROBUSTNESS OF PHYLOGENIES ACROSS ALL NODES

Here we show the results of RF scores of the evolutionary trees estimated from all pairwise distances of models and not just the leaf nodes. Due to how evolutionary trees are estimated, each node is assigned a leaf node and hence RF scores will be naturally higher than using leaf tips alone.

In Table 3 we show the table analogous to Table 1 but this time shows the results for estimating a tree using all the nodes and not just leaves.

Note that for the single tree estimate, we find much higher RF scores than in for the tree that makes just the leaves. For the consensus trees estimated from the different layers we see much smaller RF distances, only somewhat larger than the analysis of using only the leaf nodes. Note that in this scenario the $Match\%$ numbers are much smaller suggesting that the variety of trees is much larger.

Also note that that the random RF $<$ Consensus RF is still $< 0.05$ meaning that the concensus trees are reliable estimates of the underlying true topology.

Table 3: Average values over the 50 experiments.

| # Leaves | # Nodes | n | Metric | Total Weight RF (SD) | Consensus Weight RF (SD) | Match (%) | Random RF $<$ Consensus RF |
|---|---|---|---|---|---|---|---|
| 3 | 6 | 4 | Correlation | 4.25 (1.71) | 0.00 (0.00) | 46.0 | 0.000 |
| 3 | 6 | 4 | Cosine | 4.50 (1.73) | 0.00 (0.00) | 41.6 | 0.000 |
| 3 | 6 | 4 | Threshold | 5.25 (0.96) | 0.00 (0.00) | 28.1 | 0.000 |
| 3 | 6 | 4 | $L_1$ | 5.00 (1.15) | 0.00 (0.00) | 60.9 | 0.000 |
| 3 | 6 | 4 | $L_2$ | 5.00 (1.15) | 0.00 (0.00) | 60.5 | 0.000 |
| 4 | 8 | 1 | Correlation | NA | NA | 31.3 | 0.000 |
| 4 | 8 | 1 | Cosine | NA | NA | 32.1 | 0.000 |
| 4 | 8 | 1 | Threshold | NA | NA | 16.0 | 0.000 |
| 4 | 8 | 1 | $L_1$ | NA | NA | 19.1 | 0.000 |
| 4 | 8 | 1 | $L_2$ | NA | NA | 21.4 | 0.000 |
| 4 | 9 | 1 | Correlation | 10.00 (1.41) | NA | 38.9 | 0.000 |
| 4 | 9 | 1 | Cosine | 11.00 (1.41) | NA | 39.7 | 0.000 |
| 4 | 9 | 1 | Threshold | 9.00 (0.00) | NA | 42.7 | 0.000 |
| 4 | 9 | 1 | $L_1$ | 8.50 (0.71) | NA | 30.5 | 0.000 |
| 4 | 9 | 1 | $L_2$ | 7.50 (2.12) | NA | 29.0 | 0.000 |
| 5 | 9 | 1 | Correlation | 10.00 (1.41) | NA | 29.0 | 0.000 |
| 5 | 9 | 1 | Cosine | 11.00 (1.41) | NA | 26.7 | 0.000 |
| 5 | 9 | 1 | Threshold | 9.00 (0.00) | NA | 11.5 | 0.000 |
| 5 | 9 | 1 | $L_1$ | 8.50 (0.71) | NA | 45.8 | 0.000 |
| 5 | 9 | 1 | $L_2$ | 7.50 (2.12) | NA | 42.0 | 0.000 |
| 5 | 10 | 4 | Correlation | 13.25 (1.71) | 0.50 (1.00) | 15.8 | 0.067 |
| 5 | 10 | 4 | Cosine | 13.75 (1.26) | 0.50 (1.00) | 13.7 | 0.067 |
| 5 | 10 | 4 | Threshold | 6.00 (5.83) | 0.50 (1.00) | 9.5 | 0.067 |
| 5 | 10 | 4 | $L_1$ | 10.75 (1.50) | 0.50 (1.00) | 24.2 | 0.067 |
| 5 | 10 | 4 | $L_2$ | 11.00 (1.15) | 0.50 (1.00) | 22.5 | 0.067 |
| 5 | 11 | 11 | Correlation | 14.76 (1.36) | 0.18 (0.60) | 19.8 | 0.029 |
| 5 | 11 | 11 | Cosine | 14.65 (1.38) | 0.09 (0.30) | 18.6 | 0.029 |
| 5 | 11 | 11 | Threshold | 10.92 (2.72) | 0.09 (0.30) | 9.0 | 0.029 |
| 5 | 11 | 11 | $L_1$ | 12.24 (1.44) | 0.09 (0.30) | 27.3 | 0.029 |
| 5 | 11 | 11 | $L_2$ | 12.11 (1.78) | 0.09 (0.30) | 26.2 | 0.029 |
| 6 | 11 | 26 | Correlation | 14.76 (1.36) | 0.42 (0.81) | 14.2 | 0.037 |
| 6 | 11 | 26 | Cosine | 14.65 (1.38) | 0.42 (0.81) | 13.9 | 0.037 |
| 6 | 11 | 26 | Threshold | 10.92 (2.72) | 0.38 (0.75) | 9.2 | 0.037 |
| 6 | 11 | 26 | $L_1$ | 12.24 (1.44) | 0.38 (0.70) | 20.0 | 0.043 |
| 6 | 11 | 26 | $L_2$ | 12.11 (1.78) | 0.35 (0.69) | 19.1 | 0.037 |

