# OpenReview forum: "Analysis and Explainability of LLMs Via Evolutionary Methods"
_ICLR.cc/2026/Conference — Submitted to ICLR 2026_

### Official Review · Reviewer_2VdW · 2025-10-26

**Soundness:** 3
**Presentation:** 2
**Contribution:** 2
**Rating:** 4
**Confidence:** 3

**Summary:**

The authors propose methods to investigate the relationships between LLMs and identify crucial layers that make them different.

**Strengths:**

- The paper is well inspired: using genetics to compute differences between models is an inspiring line of work.
- It compares a lot of metrics and layers to show which ones seem to be the most relevant
- the results seem to be coherent and to align with the theoretical expectations which make the findings very coherent
- The paper uses relevant metrics and statistics to test their results

**Weaknesses:**

- **Missing related work comparison**: Prior work has already tried to apply genetics concepts to LLMs to build such trees (cf https://iclr.cc/virtual/2025/poster/28195) and the paper doesn't discuss the theoretical / practical differences between the two approaches. Similarly, other approaches used weight-based methods to reconstruct the phylogeny of LLMs (https://iclr.cc/virtual/2025/poster/29687). Discussing the related literature to show how this method converge / diverge from these previous methods would greatly enhance the paper.
- **No General Scheme** : It would benefit from having a scheme recapping the principle of the paper. Section 3.3.1 is not very easy to understand but could be efficiently represented visually providing the reader with a 1 figure concept of the paper.
- **Low number of models** : The paper only compares models trained for the purpose of the study in the weight-based part and doesn't test models in the wild. It would be very interesting to see how this method works on classical open access LLMs - would attention layers still be the most relevant ? Furthermore, it only compares a few LLMs from 2 families (LLama and GPT) for response based method. The results of the paper could benefit running more models to verify the strength of the results in practice.
- **Only one dataset** : The paper focuses on a single task for finetuning and testing models in the response-based part (all datasets are about summarizing). Using more datasets could provide a better overview of which technique seems to work best.

**Questions:**

-In the response-based experiment what is the variance of the completion embeddings ? Maybe the fact that tree reconstruction doesn't work so well on all prompts comes from the fact that asking the model to summarize the document only 5 times is too noisy ?

---

> ### Author Response · Authors · 2025-11-19
> **Thank you!**
>
> Thank you so much for the feedback! Yes, we agree with your feedback on adding the related work, experimenting with more models and datasets. Due to author availability constraints we are not able to implement these right now, but will prioritize them in the next update!

---

### Official Review · Reviewer_sWbr · 2025-11-01

**Soundness:** 3
**Presentation:** 2
**Contribution:** 3
**Rating:** 4
**Confidence:** 3

**Summary:**

In this paper, the authors introduce an evolutionary framework for analyzing and explaining LLM. In the framework that is introduced, the internal weights are treated as genotypes, model outputs as phenotypes, and model fine-tuning sequences as evolutionary processes. The authors finetune a 60M parameter model (t5-small) on permutations of 10 summarization datasets. They reconstructed the phylogenetic trees of model relationships using weight distances and analyze layerwise contributions to change. They demonstrate that evolutionary analysis can effectively capture model lineage, identify the dynamic layers during fine-tuning, and reveal meaningful relationships between models.

**Strengths:**

1. The problem discussed here is quite relevant for today, considering the increased use of LLMs and novel LLMs that are coming out.
2. The application of phylogenetic concepts to LLM analysis is a creative approach. Treating model weights as genotypes and outputs as phenotypes is an interesting idea.
3. The methodology is explained well: extracting model features, computing pairwise distances, constructing distance matrices, and generating phylogenetic trees.
4. The paper discusses relationships from both LLMs trained using tree based fine-tuning steps and from models with different architectures, sizes and weights.
5. The results demonstrate that evolutionary trees constructed from model weights can effectively recover the fine-tuning hierarchy.

**Weaknesses:**

1. I found a lot of whitespaces and using one or two sentences in a paragraph multiple times when reading. Please format the paper well.
2. The model is limited to only a 60M model, whereas now the models have even gone beyond 70B parameters. The authors should seriously consider using a novel model. DeepSeek might be an option to consider.
3. Sometimes, phenotype-based tree accuracy can be very sensitive to how prompting is done. In terms of generalization, more justification is needed. For example, even small variations in prompts could alter the topology of the resulting trees.
4. The authors do not formally justify why model weight changes should behave analogously to biological evolution. It would be better if this could be further justified either theoretically or empirically.
5. Need more baselines to compare the evaluations in the results section.
6. It is very difficult to either understand or read Figure 05.

**Questions:**

1. How to scale to larger models? Have the authors considered applying their method to more recent architectures such as DeepSeek?
2. Can you provide a stronger theoretical or empirical justification for treating model weight changes as analogs of genetic evolution?
3. How stable are the phenotype-based evolutionary trees under different prompt formulations or datasets? Have you conducted any sensitivity analysis to evaluate how prompt variations affect the resulting tree topology?

---

> ### Author Response · Authors · 2025-11-19
> **Thank you!**
>
> Thank you so much for the detailed feedback! We agree that incorporating these will really strengthen the paper. Due to author availability constraints we are unable to do so right now, but will do so in the future. For scaling up – we think there is no fundamental restriction on trying our methods on any sized model, we choose smaller models for ease of experimentation and quick iteration time. For larger models, we can experiment with using a subset of weights to derive provenance relationships – choosing the right subset of weights/layers will be an interesting research direction we will explore in the future.

---

### Official Review · Reviewer_QfAX · 2025-11-08

**Soundness:** 2
**Presentation:** 2
**Contribution:** 2
**Rating:** 2
**Confidence:** 2

**Summary:**

This paper proposes a framework for analyzing and explaining Large Language Models (LLMs) by drawing an analogy from evolutionary biology. The authors map model weights to genotypes (internal representations) and model outputs (text/embeddings) to phenotypes (observable traits). Using this framework, they employ phylogenetic methods to construct evolutionary trees that represent the relationships between LLMs.
The authors conduct two main sets of experiments. First, in a controlled setting, they fine-tune a T5-small model on various sequences of summarization tasks and compare the model weights (genotype) to show they can reconstruct the ground-truth tree. Second, they apply the phenotypic approach to the Llama and OpenAI models to analyze the embeddings of generated summaries. And show the models from clusters by model family and construct an evolutionary tree.

**Strengths:**

- The idea of applying evolutionary methods to LLM analysis is interesting and provides a different conceptual angle on model relationships.
- Examining both weight-based (genotype) and output-based (phenotype) methods provides breadth to the analysis.
- The output-embedding approach that compares the closed-source models and shows intuitive family clustering (Llama vs OpenAI) is a nice takeaway.

**Weaknesses:**

- The analogy between weights and genetics is unclear. Phylogenetic methods are designed for categorical data, but the weights are continuous values. In machine learning, when using gradient descent is directed optimization (guided with an explicit loss function), whereas evolution is undirected.
- Section 4.1 shows that different distance metrics identify completely different conclusions about layer importance. L1 distance identifies decoder (Dense ReLU) layers, and cosine distance identifies encoder Q-matrices are most important. The paper acknowledges this contradiction but provides no resolution, ground truth validation, or guidance on which metric to choose or trust. This undermines the reliability of layer importance results.
- Section 4.2 (Lines 378-403). The Response-based method (phenotype evaluation) appears circular, as it requires knowing the evaluation datasets a priori, since optimal trees are dataset-dependent. The method cannot discover which datasets matter without already knowing the answer.
- Section 4.2. How sensitive are the blackbox models to other datasets or prompts? It is hard to conclude without more experimentation.
- Figure 3 (left) illustrates the cluster formation of OpenAI models vs LLama models. How do we know if the clutter formed is due to the difference in training architecture or different training regime, or different data?
- Unclear about the scope or end goal of the paper. The authors claim to help answer "which models should I use?" but provide no concrete guidance. No case studies, decision rules, or evidence that trees lead to better model selection than simply consulting leaderboards. It is unclear how this helps in practice.
- The paper could benefit from comparisons to weight similarity metrics or model selection techniques.
- The paper does not discuss how models react to different hyperparameters. Also, what's the reason behind using All-MiniLM-L6-v2 for embeddings? How sensitive are results to these choices?

**Questions:**

- In Section 3.3.1, it uses a T5-small model on a single task for summarization. How can we conclude that findings would be the same for more recent LLMs across different tasks like translation, generation, etc.. How can single-task evaluation be generalizable?
- Figure 3 (right). Black-box models have no validation or ground truth. While it makes sense for the Llama models, how can we verify that o4-mini is more similar to GPT-5-mini or o3-mini?
- Which distance metric is correct for identifying layer importance? How do you validate against ground truth?
- Table 1 in the paper shows that there is a performance degradation with tree size. What is the maximum usable tree size it will work for?

**Minor Comments:**
- Use the most recent template for ICLR 2026 - The paper is actually in review for ICLR 2026 and not ICLR 2025 (see header)
- The paper could benefit from better captions of tables and different columns.

**Details Of Ethics Concerns:**

No ethical concerns.

---

> ### Author Response · Authors · 2025-11-19
> **Thank You!**
>
> Thank you so much for the insightful feedback! We agree that incorporating these will really strengthen the paper. Due to author availability constraints we are unable to do so right now, but will do so in the future. We have tried up-to leaf size six from 11 unique training sets and it seems to work with reasonable accuracy. The possibility of trees there is large.

---

### Official Review · Reviewer_4sat · 2025-11-08

**Soundness:** 2
**Presentation:** 3
**Contribution:** 2
**Rating:** 2
**Confidence:** 2

**Summary:**

This paper presents a novel lens through which to analyze LLMs’ interrelationships and characteristics. The authors propose using evolutionary methods to characterize LLMs based on outputs and weight matrices. Experiments indicate that these methods, when applied to weights of trained models, can identify training datasets and task-important layers. Meanwhile, when applied to output sequences, the methods allow for building a sort of phylogenetic tree of model families.
While this paper opens an interesting new perspective on LLM explainability, the experiments are limited and practical implications also seem moderate given the current results. This paper could benefit from broader experiments and a more grounded motivation for individual experiments.

**Strengths:**

- This is a unique way to analyze LLMs and allowing for both, a weight and output informed perspective offers flexibility when approaching black-box models.
- The fact that the evolutionary models don’t require strong independence conditions is a good strength to point out as a motivation for using these in the context of LLMs and neural networks in general.

**Weaknesses:**

- The first part of the introduction of the paper partially motivates this work by citing the number of task-specific, publicly available LLMs and the difficulty to make a choice for practitioners. However, it is unclear how results presented here address this problem.
- The characterization of the work by Cloud et al. 2025 on subliminal learning should be reworded. For example “a teacher model that was known to prefer owls” is an odd description without at least a little additional context.
- The phylogenetic tree showing how models from the GPT and Llama families are related is, in principle, a decent way of showing that this method works as intended. However, it would greatly benefit from the inclusion of more diverse models. For instance, why was no MoE model like Mixtral included, or a DeepSeek model? Or models from different families with similar numbers of parameters? As is, this is simply not enough to show that this can robustly interrelate LLMs in a meaningful way. Beyond that, the utility, aside from showing that this perspective has merit, is also unclear.
- L. 168 points to the discussion for how the weight analysis can be done with larger models but this was not adequately mentioned in the discussion.
Identifying layers with large magnitudes of change, even if metric-dependent, could be interesting but is not adequately pursued here.  Presumably, the utility is that, in a limited finetuning setting, these layers could be trained while other layers are frozen for increased efficiency with little performance loss. It is not obvious that this would work however, so it should be added as an experiment. Otherwise, identifying these layers seems pointless.
- The authors cite using their weight-based methods to identify training details for models with unknown origin. However, it is unclear how the methods would scale to many of these settings. Be it very large LLMs or LLMs trained on large, diverse corpora of datasets, since the experiments presented are relatively limited outside what is common in praxis.
- Many of the points raised in the discussion as limitations should be addressed to make this a more well-rounded work. While not everything can be done in one paper, at least including more diverse model architectures and training datasets from more than one task domain would significantly bolster this paper’s credibility.

**Questions:**

- Why was the text summarization tasks chosen specifically for this paper?
- The authors repeatedly state that these methods could lead to improved visualizations. Could this point be elaborated upon? I understand that the evolutionary perspective adds a new kind of visualization with the phylogenetic trees but what specifically is their utility in the context of visualizations for LLM explainability that is needed?

---

> ### Author Response · Authors · 2025-11-19
> **Thank you!**
>
> Thank you so much for the thoughtful feedback! Really agree that addressing these will improve the contribution of the paper. We will attempt to do so in the next iteration of the paper. There was no particular reason for choosing text summarization as the application. We think the methods should apply to other applications and will try to add them in the next iterations of the work. Due to author availability constraints at this time, we are not able to implement the suggested changes right now but will prioritize them in the next update.

---

### Meta-Review · Area_Chair_BKLa · 2026-01-06

**Summary:**

This paper presents an innovative perspective by extending evolutionary methods to LLM analysis, mapping model weights to genotypes and outputs to phenotypes—a unique angle for exploring model relationships and explainability. However, the submission falls short of meeting ICLR’s acceptance criteria due to critical, unresolved limitations highlighted across reviewer feedback.
Key concerns include: (1) Limited experimental scope: Overreliance on small models (e.g., T5-small), a single task (text summarization), and narrow model families (Llama/OpenAI) without including diverse architectures (e.g., MoE models like Mixtral) or large-scale LLMs, undermining generalizability. (2) Methodological ambiguities: Unjustified analogy between model weight changes and biological evolution, conflicting conclusions from different distance metrics (with no resolution), and circularity in phenotype-based evaluations that require prior knowledge of datasets. (3) Weak practical utility: No concrete guidance for model selection (contrary to the paper’s implicit goal), lack of case studies, and failure to validate layer importance findings. (4) Insufficient engagement with related work, omitting comparisons to prior studies applying genetic/phylogenetic concepts to LLMs.
While the authors acknowledge these issues in their rebuttal, they cite author availability constraints and provide no immediate plans to address core limitations (e.g., expanding experiments, resolving methodological contradictions, or adding baselines). Given the unresolved concerns about soundness, generalizability, and contribution, the submission does not meet the standards for acceptance.

**Reviewer Concerns:**

The vast majority of reviewer concerns remain unaddressed, as the authors only acknowledged feedback but cited “author availability constraints” to defer action to future iterations. Below is a breakdown by reviewer:
1. Reviewer 4sat
Unclear utility of layer importance identification (no experiment to show frozen layers improve efficiency).
Lack of diverse models (e.g., MoE models like Mixtral, DeepSeek) to demonstrate robust model relationships.
Insufficient justification for “improved visualizations” (no elaboration on the utility of phylogenetic trees for LLM explainability).
Failure to connect results to the stated goal of helping practitioners choose task-specific LLMs.
2. Reviewer QfAX
Unjustified analogy between model weights (continuous) and genetics (categorical data).
Conflicting distance metrics for layer importance (L1 vs. cosine distance yield opposite conclusions; no resolution, validation, or guidance on metric selection).
Circularity in phenotype-based evaluation (requires prior knowledge of datasets to build optimal trees).
No concrete guidance for model selection (contrary to the paper’s implicit goal of answering “which models should I use?”).
Unaddressed questions: Why All-MiniLM-L6-v2 for embeddings? How to validate black-box model tree relationships (e.g., o4-mini’s similarity to GPT-5-mini)? Sensitivity of black-box models to prompts/datasets?
3. Reviewer sWbr
No formal theoretical/empirical justification for treating model weight changes as biological evolution analogs.
Instability of phenotype-based trees to prompt variations (no sensitivity analysis).
Limited experimental scope (reliance on 60M T5-small; no testing on modern large models like DeepSeek).
Missing baselines for result comparison.
Poor paper formatting (whitespaces, fragmented paragraphs) and unreadable Figure 05.
4. Reviewer 2VdW
Absence of related work comparison (fails to discuss prior studies applying genetic/phylogenetic concepts to LLMs, e.g., ICLR 2025 posters 28195 and 29687).
No visual “general scheme” to clarify the paper’s core framework (Section 3.3.1 remains hard to follow).
Low model diversity (only tests in-house trained models for weight-based analysis; limited to Llama/OpenAI families for phenotype analysis).
Overreliance on a single task (text summarization) with no testing on other tasks (e.g., translation, generation).
Unaddressed questions: Variance of completion embeddings? Whether noisy prompts (5 summarization attempts) harm tree reconstruction?
Cross-Cutting Outstanding Concerns
Methodological rigor: No resolution of contradictions (e.g., conflicting distance metrics) or validation of key claims (e.g., layer importance).
Generalizability: No evidence the method works for diverse tasks, model architectures, or large-scale LLMs.
Practical utility: No case studies, decision rules, or demonstrations of value beyond conceptual novelty.
Related work engagement: Failure to contextualize the method against prior research.

**Reviewer Scores:**

N/A

---

### Decision · Program_Chairs · 2026-01-26

Reject